# Long-term uncertainty quantification in WRF-modeled offshore wind resource off the US Atlantic coast

Nicola Bodini[1], Simon Castagneri[1], and Mike Optis[2]

[1]National Renewable Energy Laboratory, 15013 Denver W Pkwy, Golden, CO, 80401, USA
[2]Veer Renewables Inc., Courtenay, British Columbia, V9N 9B4, Canada

**Correspondence:** Nicola Bodini (nicola.bodini@nrel.gov)

**Abstract.** Uncertainty quantification of long-term modeled wind speed is essential to ensure stakeholders can best leverage wind resource numerical data sets. Offshore, this need is even stronger given the limited availability of observations of wind speed at heights relevant for wind energy purposes and the resulting heavier relative weight of numerical data sets for wind energy planning and operational projects. In this analysis, we consider the National Renewable Energy Laboratory's 21-year updated numerical offshore data set for the U.S. East Coast and provide a methodological framework to leverage both floating lidar and near-surface buoy observations in the region to quantify uncertainty in the modeled hub-height wind resource. We first show how using a numerical ensemble to quantify the uncertainty in modeled wind speed is insufficient to fully capture the model deviation from real-world observations. Next, we train and validate a random forest to vertically extrapolate near-surface wind speed to hub height using the available short-term lidar data sets in the region. We then apply this model to vertically extrapolate the long-term near-surface buoy wind speed observations to hub height so that they can be directly compared to the long-term numerical data set. We find that the mean 21-year uncertainty in 140 m hourly average wind speed is slightly lower than $3~\mathrm{m\,s^{-1}}$ (roughly 30% of the mean observed wind speed) across the considered region. Atmospheric stability is strictly connected to the modeled wind speed uncertainty, with stable conditions associated with an uncertainty which is, on average, about 20% larger than the overall mean uncertainty.

*Copyright statement.* This work was authored in part by the National Renewable Energy Laboratory, operated by Alliance for Sustainable Energy, LLC, for the U.S. Department of Energy (DOE) under Contract No. DE-AC36-08GO28308. Funding was provided by the U.S. Department of Energy Office of Energy Efficiency and Renewable Energy Wind Energy Technologies Office. Support for the work was also provided by the National Offshore Wind Research and Development Consortium under Agreement no. CRD-19-16351. The views expressed in the article do not necessarily represent the views of the DOE or the U.S. Government. The U.S. Government retains and the publisher, by accepting the article for publication, acknowledges that the U.S. Government retains a nonexclusive, paid-up, irrevocable, worldwide license to publish or reproduce the published form of this work, or allow others to do so, for U.S. Government purposes.

# 1 Introduction

The offshore wind energy industry has been growing at an unprecedented pace worldwide (Musial et al., 2022). While the United States currently only have 42 MW of installed offshore wind capacity (Global Wind Energy Council, 2023), many

more turbines are planned to be built in the coming years, with a target of at least 30 GW of installed capacity by 2030 (Room, 2021). With a total offshore technical resource potential thought to be about twice the current national energy demand (Musial et al., 2016), offshore wind energy represents a valuable clean source of energy to meet future needs. Such growth requires the existence of accurate long-term wind resource data sets to help interested stakeholders in their preconstruction energy evaluations (Brower, 2012). Given the technical, logistical, and economical challenges in deploying instruments capable of

characterizing the offshore wind resource at heights relevant for wind energy purposes, numerical weather prediction (NWP) models are often used to provide continuous (in space and time), high-resolution wind resource assessment. The National Renewable Energy Laboratory (NREL) recently released a state-of-the-art offshore wind resource assessment product based on 21-year-long simulations using the Weather Research and Forecasting (WRF) model (Skamarock et al., 2019) for all U.S. offshore waters. This updated data set is intended to replace the offshore component of the WIND Toolkit (Draxl et al., 2015).

Given the high stakes at play connected to the planned future growth of offshore wind energy, it is essential that data sets such as NREL's quantify and communicate the uncertainty that comes with the modeled wind resource. In fact, previous studies showed how even a small uncertainty change in the modeled mean wind speed translates into an almost double uncertainty for the long-term prediction of the annual energy production of a wind plant (Johnson et al., 2008; White, 2008; Holstag, 2013; Truepower, 2014), which is associated with significantly higher interest rates for new wind project financing.

A somewhat conventional approach to quantify uncertainty from NWP models is to consider the variability of the quantity of interest – in our case wind speed – across a number of numerical ensemble members, which are different realizations of the numerical model obtained by tweaking the numerical model setup. Many different setup choices can affect the wind speed predicted by an NWP model: which planetary boundary layer (PBL) scheme to adapt in the simulations (Ruiz et al., 2010; Carvalho et al., 2014a; Hahmann et al., 2015; Olsen et al., 2017), which large-scale atmospheric product to use to force

the model runs (Carvalho et al., 2014b; Siuta et al., 2017), the model horizontal resolution (Hahmann et al., 2015; Olsen et al., 2017), the model spin-up time (Hahmann et al., 2015), and data assimilation techniques (Ulazia et al., 2016) are some of the main contributing factors to wind speed variability across different model runs. Running a numerical ensemble can quantify an ensemble-derived uncertainty, and Bodini et al. (2021) showed how using machine learning approaches can reduce the temporal extent of the computationally expensive ensemble runs necessary to quantify this type of uncertainty (called

"boundary condition and parametric uncertainty" in their article) over a long-term period.

However, quantifying only the uncertainty connected to the possible choices in model setup presents several limitations. In fact, the magnitude of the uncertainty that can be quantified from the NWP ensemble variability is strictly connected to the limited number of choices sampled within the considered model setups. NWP model ensembles tend to lead to an underdispersive behavior (Buizza et al., 2008; Alessandrini et al., 2013), so that only a limited component of the actual wind speed

error with respect to observations can be quantified from them. The proper, full uncertainty in NWP-model-predicted wind

speed can only be quantified when leveraging direct observations of the wind resource, concurrent with the modeled period. In this ideal scenario, the residuals between modeled and observed wind speed can be calculated, and the model error can be quantified both in terms of its bias (i.e., the mean of the residuals) and uncertainty (or, in simple terms that will be refined later in the paper, the standard deviation of the residuals).

It is important to remember that, while observational data sets are essential for a proper quantification of numerical uncertainty, they also come with their own uncertainty, which should therefore be considered in the overall numerical uncertainty quantification. Observational uncertainty stems from a variety of factors (Yan et al., 2022). The size and representativeness of the available observations is a primary factor to keep in mind, especially when less than one year of data is available for use, and the seasonal cycle of atmospheric variables cannot be accurately captured. Also, observations come with inherent instrumental

uncertainty, whose general guidelines are well described in the ISO Guide to the expression of Uncertainty in Measurements (JCGM 100:2008, 2008b, a), often referred to as "GUM". Specific considerations will then apply for each specific instrument, so that for example lidars will have different uncertainties compared to traditional anemometers.

In our analysis, we present and validate a novel methodology to assess long-term (in our case, 21-year) uncertainty quantification for modeled wind speed in the mid-Atlantic region of the United States by leveraging available observations of offshore

wind. While in our analysis we focus on the U.S. mid-Atlantic domain, the methodology could be applied in other offshore regions as well. In Sect. 2 we describe our long-term WRF simulations, and the lidar and buoy observations that we leverage to assess uncertainty in the modeled data set. In Sect. 3 we present our proposed methodology to assess long-term uncertainty in modeled wind speed by comparing it with vertically extrapolated observed winds. In Sect. 4 we dive deeper into the already mentioned topic of using numerical ensembles to quantify uncertainty and provide a demonstration of the limits of such an

approach. We accurately validate our uncertainty quantification approach in Sect. 5, present the main results of our long-term uncertainty quantification (in terms of uncertainty in hourly average wind speed) in Sect. 6, and conclude our analysis in Sect. 7.

## 2    Data

### 2.1    Numerical data

We use NREL's WRF-modeled long-term wind speed data in the mid-Atlantic region (Bodini et al., 2020). The model is run from January 2000 to December 2020, at 2-km horizontal resolution, 5-minute temporal resolution, with nine vertical levels in the lowest 200 m, using the model setup illustrated in detail in Table 1. Multiple model setups (obtained by tweaking the reanalysis forcing, PBL scheme, sea surface temperature (SST) product, and land surface model) were considered, and the model setup described here was chosen, as it best validated against available lidar observations in the region (Pronk et al.,

2022). The WRF simulations are run separately for each month and then concatenated into a single, 21-year time series at each location. We use a 2-day spin-up period at the beginning of each simulated month (e.g., July simulations started on 29 June) to allow the model to develop sufficiently from the initial conditions and stabilize. We apply atmospheric spectral nudging to

the outer domain every 6 hours, and find that the accuracy of simulated winds is not impacted by the length of the 1-month simulation periods (i.e., the model errors at the beginning of each month are not lower than at the end of the month, on average).

**Table 1.** Key attributes of the 21-year WRF simulations used in this study.

| Feature | Specification |
| --- | --- |
| WRF version | 4.2.1 |
| Grid spacing | 6 km, 2 km (nested) |
| Output time resolution | 5 minutes |
| Vertical levels | 61 |
| Near-surface-level heights (m) | 12, 34, 52, 69, 86, 107, 134, 165, 200 |
| Atmospheric forcing | ERA-5 reanalysis (Hersbach et al., 2020) |
| Planetary boundary layer scheme | Mellor–Yamada–Nakanishi–Niino Level 2.5 (Nakanishi and Niino, 2009) |
| Land surface model | Noah (Ek et al., 2003) |
| Microphysics | Ferrier (Schoenberg Ferrier, 1994) |
| Longwave radiation | Rapid radiative transfer model (Mlawer et al., 1997) |
| Shortwave radiation | Rapid radiative transfer model |
| Topographic database | Global multiresolution terrain elevation data from the |
| | U.S. Geological Survey and National Geospatial-Intelligence Agency |
| Land-use data | Moderate Resolution Imaging Spectroradiometer 30 s (Justice et al., 2002) |
| Cumulus parameterization | Kain–Fritsch (6 km domain) (Kain and Fritsch, 1993) |
| Sea surface temperature product | Operational Sea Surface Temperature |
| | and Sea Ice Analysis (OSTIA) (Donlon et al., 2012) |

## 2.2 Observations

An ideal uncertainty quantification over the 21-year extent of our offshore wind resource numerical data set would require concurrent 21-year time series of observed winds at a height relevant for wind energy purposes and at as many locations as possible within the modeled domain. In reality, such extensive observations do not exist. We therefore consider two sets of observations and apply a machine-learning-based approach to leverage the advantages of each. On one hand, we use lidar observations in the region, which provide measurements at hub height but only over a handful of months. On the other hand, we consider observations from National Data Buoy Center (NDBC) buoys, which are available over much longer time periods but only provide observations close to the sea surface.

### 2.2.1 Lidar observations

We consider four sets of lidar measurements taken from three lidars in the region (Fig. 1):

- The New York State Energy Research and Development Authority (NYSERDA) E05 North data set (OceanTech Services/DNV GL, 2020), collected by a ZephIR ZX300M unit, from 12 August 2019 to 19 September 2021. Most observations from the lidar and other instruments on the lidar buoy are provided as 10-minute averages, after proprietary quality checks are applied to the data. We use wind speed and wind direction, which are available at 3.1 m and then every 20 m from 20 m to 200 m above sea level, and air temperature. Sea surface temperature is provided as hourly average values.

- The NYSERDA E06 South data set, collected by a second ZephIR ZX300M unit, from 4 September 2019 to 27 March 2022. The same data considerations listed above for the E05 instrument apply to this unit as well. For this unit, data availability statistics, as defined by the proprietary quality controls applied to the instrument, were released, and show that the lidar data availability decreases with height from 83 % to 76 %, while near-surface measurements have an availability greater than 96 %.

- The Atlantic Shores consortium 06 data set, collected by a third ZephIR ZX300M unit, from 26 February 2020 to 14 May 2021. Data (wind speed and wind direction profiles, air temperature, and sea surface temperature) are available at a 10-minute resolution. Wind speed and direction data are provided at 4.1 m, all 20 m intervals from 40 to 200 m, and 250 m above sea level.

- The Atlantic Shores consortium 04 data set, collected by the same unit, which was moved to a different location and recorded data from 14 May 2021 to 6 March 2022, with the same data specifications as the other Atlantic Shores data set.

Some of the considered floating lidar platforms were not operational for part of their overall deployment period. Figure 2 shows the monthly coverage for each buoy. For all lidars, we calculate hourly averages of all the relevant variables. If a variable is missing from some 10-minute periods, the hourly average value is still calculated using the available data within that 60-minute period. We kept only hourly time stamps where hourly average values of 140 m wind speed, near-surface wind speed, near-surface wind direction, air temperature, and sea surface temperature were all available. Table 2 shows the mean 140-m wind speed from the four lidar data sets, calculated using the selected time stamps as described here.

**Table 2.** Mean 140-m mean speed from the four lidar data sets (after all the quality checks described in Section 2.2.1 were applied).

| Lidar | Mean 140-m wind speed (m/s) |
| --- | --- |
| NYSERDA E05 North | 9.38 |
| NYSERDA E06 South | 9.99 |
| Atlantic Shores 04 | 9.44 |
| Atlantic Shores 06 | 9.69 |

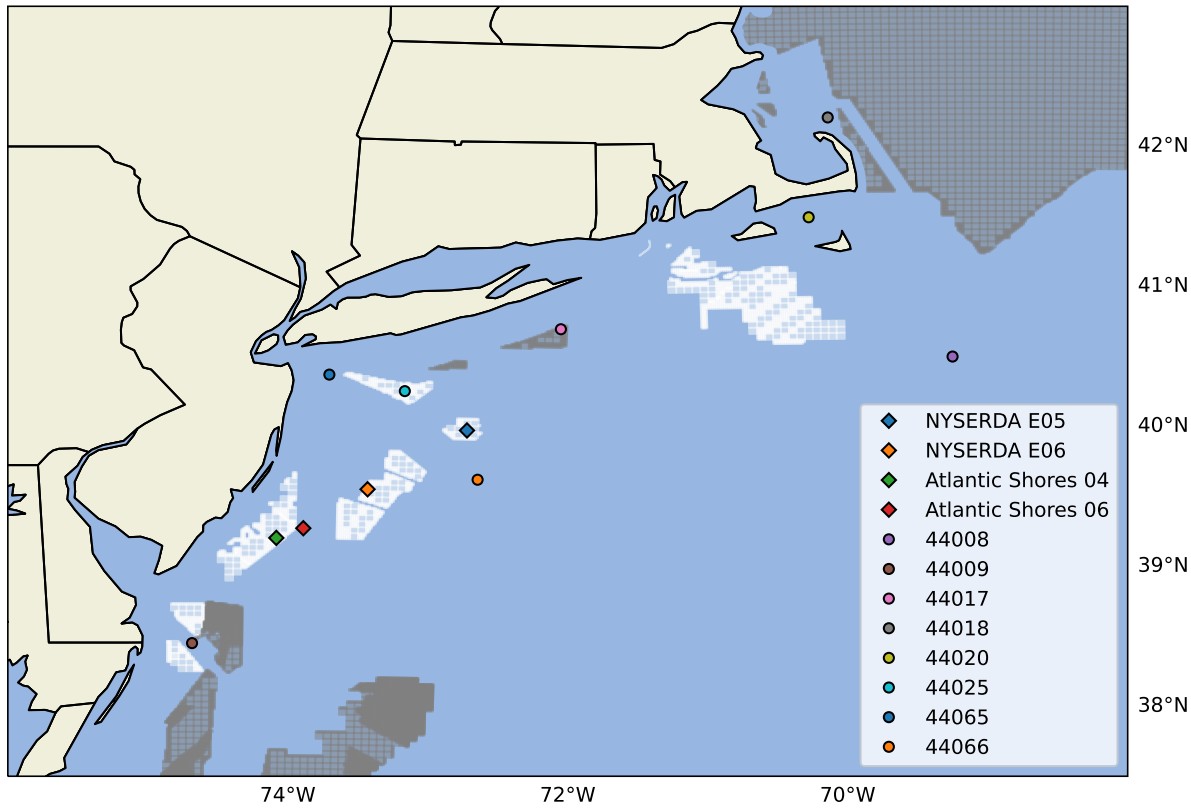

**Figure 1.** Map of the observational data sets used in the analysis. Lidar locations are shown as diamonds, NDBC buoys are shown as dots. Wind lease areas are shown in white, wind planning areas in gray. The distance between the two NYSERDA lidars is about 75 km, the two Atlantic Shores lidars are about 20 km from each other, and finally the distance between the NYSERDA E05 and Atlantic Shores 04 lidars is about 145 km.

### 2.2.2 NDBC buoy observations

Finally, we consider long-term near-surface observations from eight buoys managed by the NDBC (locations in Fig. 1). At
125  each buoy, we consider observations of air and sea surface temperatures, and wind speed and direction. Table 3 shows the
heights at which each variable is recorded. One buoy (ID 44009) provides observations at slightly different heights than all the
other buoys, but we determined that this minor difference would have a minimal impact on our results. Whenever available, we
take data from the full 21-year period that is modeled in our WRF runs. If the full 21-year period is not available, we consider
observations from the start of each buoy's period of record to the end of 2020. Data are provided at 10-minute resolution for the
130  most recent years, and 1-hour resolution for the first few years at the beginning of the century. To be consistent, we calculate
1-hour averages across the whole 21-year period. As done for the lidar data, if a variable is missing from some 10-minute

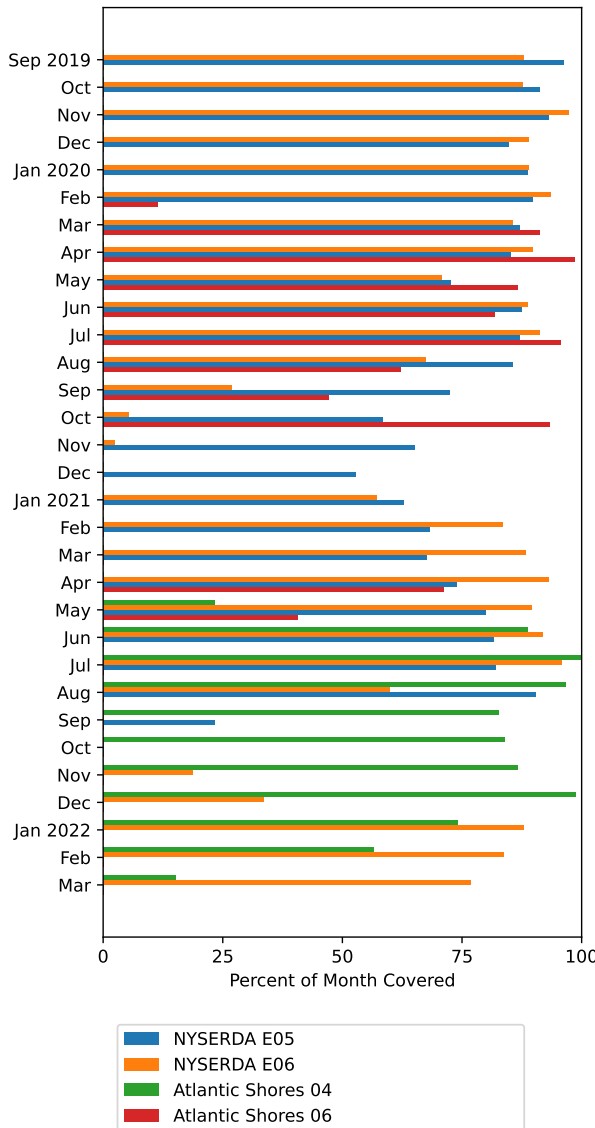

**Figure 2.** Data availability chart for the four lidar data sets. We kept only hourly time stamps for which we could calculate hourly average values for all the variables considered in this analysis, as detailed in the text.

intervals, the hourly average value is still calculated using the available data within that 60-minute period. Only hourly time stamps where we have valid hourly average values for all the relevant variables are kept for the analysis.

**Table 3.** List of NDBC buoys used in this analysis.

| Name | Wind speed height (meters above water line) | Air temperature height (meters above water line) | Sea surface temperature height (meters below water line) | Period of record used |
|---|---|---|---|---|
| 44008 | 4.1 | 3.7 | 1.5 | 2000–2020 |
| 44009 | 3.8 | 3.4 | 2.0 | 2000–2020 |
| 44017 | 4.1 | 3.7 | 1.5 | 2002–2020 |
| 44018 | 4.1 | 3.7 | 1.5 | 2002–2020 |
| 44020 | 4.1 | 3.7 | 1.5 | 2009–2020 |
| 44025 | 4.1 | 3.7 | 1.5 | 2000–2020 |
| 44065 | 4.1 | 3.7 | 1.5 | 2008–2020 |
| 44066 | 4.1 | 3.7 | 1.5 | 2009–2020 |

## 3 Methods

To be able to leverage the long-term time series of the NDBC buoys for an uncertainty quantification that is relevant to offshore wind energy purposes, the buoy observations need to be vertically extrapolated to a height of interest for commercial wind energy development. Several techniques exist to vertically extrapolate wind speeds. Traditional approaches include using a power law relationship (Peterson and Hennessey Jr, 1978) or a logarithmic profile more firmly based on the Monin–Obukhov Similarity Theory (Monin and Obukhov (1954)). However, recent research has shown how machine-learning-based techniques outperform these conventional extrapolation approaches, both onshore (Vassallo et al., 2020; Bodini and Optis, 2020b, a) and offshore (Optis et al., 2021).

### 3.1 Machine learning algorithm for wind speed vertical extrapolation

We use a random forest machine learning model, a robust ensemble regression algorithm that has been successfully applied to similar applications. In this work, we use the `RandomForestRegressor` module in Python's Scikit-learn (Pedregosa et al., 2011). Additional details on random forests can be found in machine learning textbooks (e.g., Hastie et al. (2005)). We train the regression model to predict hourly average wind speed at 140 m. We use the following observed variables as inputs to the model, all as hourly averages:

- Near-surface wind speed

- Near-surface wind direction[1]

---

[1] To preserve the cyclical nature of this variable, we calculate and include as inputs its sine and cosine. We note that both sine and cosine are needed to identify a specific value of the cyclical variable, because each value of sine only (or cosine only) is linked to two different values of the cyclical variable. For example, the sine of wind direction is 0 for both $90°$ and $270°$, but once their (different) cosines are introduced as well, the two can be identified in a univocal way.

**Table 4.** Algorithm hyperparameters considered for the random forest, their sampled values in the cross validation, and chosen value in the final version of the model used in Sect. 6.

| Hyperparameter | Sampled Values | Selected value |
|---|---|---|
| Number of estimators | 10–20 | 16 |
| Maximum depth | 1–10 | 9 |
| Maximum number of features | 1–10 | 8 |
| Minimum number of samples to split | 2–11 | 10 |
| Minimum number of samples for a leaf | 1–15 | 6 |

- Air temperature

- Sea surface temperature

- Difference between air temperature and SST

- Time of day[1]

- Month[1].

We use a 5-fold cross validation, where we build the testing set using a consecutive 20 % of the observations from each calendar month in the period of record to ensure that the learning algorithm can be (trained and) tested on a set of data that captures the seasonal variability at each site well. Also, we consider the hyperparameter ranges shown in Table 4 and sample 20 randomly selected combinations of them during the cross-validation process. The combination of hyperparameters that leads to the lowest root-mean-square error (RMSE) between the observed and random-forest-predicted 140 m wind speed is selected and used in the final model. We note that the range of hyperparameters used limits the complexity of the random forest and the computational resources needed to train the model. We tested using a larger forest with deeper trees, but that led to overfitting the available training data.

The chosen splitting approach in the cross-validation ensures that short-term autocorrelation in the data does not artificially increase the measured skill of the algorithm (as it would happen if training and testing data sets were randomly chosen without imposing a consecutive data requirement). However, potential lag correlation in the data could still play a role. Therefore, we tested whether using a single, consecutive 20% of the data for testing leads to significantly different results in terms of model accuracy. We tested this on the two NYSERDA lidars (because they both span a period of record longer than one year, and therefore can still capture a full seasonal cycle in their training phase even when a single 20% of the data is kept aside for testing), and found no significant difference in the model performance.

## 3.2 Uncertainty quantification

As detailed in Sects. 5 and 6, we apply the random forest algorithm to vertically extrapolate wind speed up to 140 m at the location of the eight NDBC buoys. To assess the uncertainty in WRF-modeled long-term wind speed at each buoy location, we first calculate the time series of the residuals between 140 m modeled winds and 140 m extrapolated winds. Then, we calculate the average and the standard deviation of each residual time series, which represent the bias and uncertainty components of the model error at each location, respectively. Next, we compare the biases across all the measurement locations (in our case, the eight buoys):

- If the standard deviation of the biases is smaller than the typical single-site uncertainty (i.e., the average of each site's standard deviation of the residuals), then the latter is a good measure of the model uncertainty.

- If the standard deviation of the biases exceeds the typical single-site uncertainty, then the model uncertainty is dominated by the unpredictable bias and can be estimated from the standard deviation of the biases itself.

Finally, when estimating model uncertainty from measurements, it is important to remember that the measurements themselves have an uncertainty. In our case, we need to consider both the actual instrumental uncertainty ($\sigma_{\mathrm{obs}}$) and the uncertainty connected to the fact the WRF-modeled wind speed will not be compared to directly observed wind speed, but rather to wind speed that has been vertically extrapolated from near-surface observations, using a less-than-perfect algorithm, which was trained at sites different from the ones it is being applied to ($\sigma_{\mathrm{ML}}$). Both these uncertainty components are passed on to the model. For simplicity, we assume that all three components are not correlated with each other, and add $\sigma_{\mathrm{obs}}$ and $\sigma_{\mathrm{ML}}$ in quadrature to the model uncertainty $\sigma_{\mathrm{WRF}}$ estimated using the steps above, to obtain a total uncertainty quantification (JCGM 100:2008, 2008b):

$$\sigma_{\mathrm{tot}} = \sqrt{\sigma_{\mathrm{WRF}}^2 + \sigma_{\mathrm{obs}}^2 + \sigma_{\mathrm{ML}}^2} \tag{1}$$

As will be detailed in the following sections, we will use constant values across the whole considered region for both the instrumental uncertainty $\sigma_{\mathrm{obs}}$ and the extrapolation-related uncertainty $\sigma_{\mathrm{ML}}$. This means that these two uncertainty components will not directly consider the fact that the data set the WRF-modeled hub height wind speed will be compared to has gradually lower quality as we move away from the lidars used to train the extrapolation algorithm. Therefore, this aspect is folded into the uncertainty quantified by $\sigma_{\mathrm{WRF}}$, which therefore assumes a larger connotation compared to the pure WRF model uncertainty: in fact, only if the WRF-modeled wind speed was compared to the *true* hub height wind speed, $\sigma_{\mathrm{WRF}}$ would be a pure quantification only of the uncertainty in numerically modeling hub height winds. Therefore, while for this proof of concept analysis the assumption of uncorrelated uncertainty components can be considered as sufficiently reasonable, and therefore equation 1 justified, future follow-up analyses could explore the potential correlation between different uncertainty sources to further refine the quantification approach we use here.

## 4  Limits of using an ensemble-based approach for uncertainty quantification

Before diving deep into the uncertainty quantification using the approach outlined in the previous section, we are interested in confirming the limitations of using an ensemble-derived uncertainty as a way to fully capture an NWP model uncertainty, as discussed in the introduction. To do so, we run a 1-year (September 2019 to August 2020) WRF ensemble across the mid-Atlantic region, and calculate the (temporal) mean of the modeled 140 m wind speed standard deviation calculated across the ensemble at each time stamp at the location of the two NYSERDA lidars. These values quantify, in a rather simple yet often used fashion that neglects any correlation among the ensemble members, the model ensemble-derived uncertainty at the two lidar locations. We then compare these values with the total model uncertainty, calculated using Eq. (1). We compute $\sigma_{\mathrm{WRF}}$ as the standard deviation of the 1-year time series of the residuals between 140 m wind speed from the main WRF run (i.e., the one with the setup used for the full 21-year period) and concurrent observations from the two NYSERDA lidars. We assume the uncertainty in the lidar observations $\sigma_{\mathrm{obs}}$ to be 3 % of the reported lidar 140 m wind speed across the considered period following what was reported in the NYSERDA lidar documentation (OceanTech Services/DNV GL, 2020), and therefore equal to $0.31\,\mathrm{m\,s^{-1}}$. Finally, in this case, $\sigma_{\mathrm{ML}} = 0$ because we are not applying any vertical extrapolation approach. We perform both calculations from hourly average time series of modeled and observed wind speed.

For this exercise, we consider 16 ensemble members, obtained by considering all the possible combinations of setups resulting from the following four variations:

- Reanalysis forcing: We consider the state-of-the-art ERA5 reanalysis product developed by the European Centre for Medium-Range Weather Forecasts (ECMWF) (Hersbach et al., 2020) and the Modern-Era Retrospective analysis for Research and Applications, Version 2 (MERRA-2) (Gelaro et al., 2017), developed by the National Aeronautics and Space Administration (NASA). Both these reanalysis products have been widely used in applications related to wind energy and represent the most advanced reanalysis products available to date.

- Planetary boundary layer scheme: We consider the Mellor–Yamada–Nakanishi–Niino (MYNN) (Nakanishi and Niino, 2009) and the Yonsei University (YSU) (Hong et al., 2006) PBL schemes. These two models are widely considered the two most popular PBL schemes in WRF, especially when considering wind-related applications: YSU was used in the WIND Toolkit (Draxl et al., 2015), and MYNN was used in the New European Wind Atlas (Hahmann et al., 2020; Dörenkämper et al., 2020).

- Sea surface temperature product: We consider the Operational Sea Surface Temperature and Sea Ice Analysis (OSTIA) data set produced by the UK Met Office (Donlon et al., 2012) and the National Center for Environmental Prediction (NCEP) Real-Time Global (RTG) SST product (Grumbine, 2020).

- Land surface model (LSM): We consider the Noah LSM and the updated Noah-Multiparameterization (Noah-MP) LSM (Niu et al., 2011).

Table 5 summarizes the result of this comparison. We find that, while the ensemble-derived uncertainty at either lidar are lower than $1\,\mathrm{m\,s^{-1}}$ (roughly equal to 10% of the mean observed 140-m wind speed, see Table 2), the actual model uncertainty

is instead closer to $2\,\mathrm{m\,s^{-1}}$ (roughly equal to 20% of the mean observed 140-m wind speed). This comparison clearly confirms how an NWP model's uncertainty quantified from the variability across a numerical ensemble can only quantify a limited component of the full model uncertainty – in our specific case for hub-height wind speed – with a relative difference of about 50 %.

**Table 5.** Comparison between ensemble-derived uncertainty and total model uncertainty in 140 m wind speed at the locations of the two NYSERDA lidars.

| Lidar | Ensemble-based uncertainty $(\mathrm{m\,s^{-1}})$ | WRF uncertainty $(\mathrm{m\,s^{-1}})$ | Total model uncertainty $(\mathrm{m\,s^{-1}})$ |
|---|---|---|---|
| NYSERDA E05 | 0.95 | 1.90 | 1.93 |
| NYSERDA E06 | 0.96 | 1.84 | 1.87 |

## 5 Machine learning wind speed vertical extrapolation validation

Given the inappropriate uncertainty quantification resulting from a numerical ensemble, we are now ready to start working on our machine learning vertical extrapolation approach to be able to apply our proposed pipeline for a broader uncertainty quantification. For the long-term uncertainty quantification, the random forest algorithm needs to be applied at each buoy location to derive a long-term time series of extrapolated winds, which will be compared to the WRF-modeled wind resource. However, before doing so, the regression model first needs to be trained at the floating lidar sites so that it can learn how to model hub-height wind speed in the region from a set of near-surface data. Also, the generalization skill of the model needs to be quantified, as a proper uncertainty quantification needs to also account for the uncertainty of the approach used to obtain the observation-based long-term time series of hub-height winds at each buoy location.

First, we verify that the learning algorithm we have chosen does not overfit the data. To do so, we compare the training and testing RMSE at the four lidar data sets in Table 6. The fact we do not systematically see significantly larger testing RSME confirms that the random forest does not overfit the data at any of the lidar sites, so that we can continue in our validation by assessing its generalization skills.

**Table 6.** Comparison of training and testing RMSE (in modeling hourly average wind speed at 140 m) at the four lidar data sets.

| Lidar data set | Training RMSE $(\mathrm{m\,s^{-1}})$ | Testing RMSE $(\mathrm{m\,s^{-1}})$ |
|---|---|---|
| NYSERDA E05 | 1.09 | 1.14 |
| NYSERDA E06 | 1.21 | 1.18 |
| Atlantic Shores 04 | 1.17 | 1.22 |
| Atlantic Shores 06 | 1.27 | 1.33 |

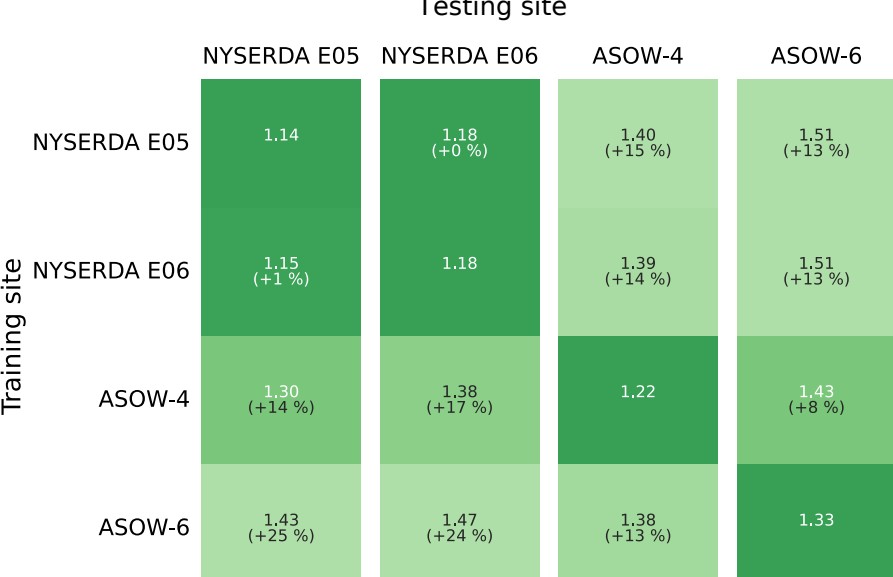

**Figure 3.** Testing root-mean-square error (in $\mathrm{m\,s^{-1}}$) in predicting hourly average wind speed at 140 m above sea level for the different lidar data sets, as a function of the data set used to train the random forest.

250 We validate the machine learning extrapolation model using a "round-robin" approach. In fact, it is neither fair nor practically useful to assess the skill of the regression algorithm when it is trained and tested at the same lidar location, as that is not our actual application of the model. Instead, one should assess the performance of the extrapolation approach when the random forest is trained at one lidar and then used to extrapolate wind speed at a different lidar, where the model has no prior knowledge (or, better yet, limited prior knowledge since the training site is still in the vicinity) of the wind conditions at the site. Figure 3

255 shows the result of such a round-robin validation; we compare the RMSE of the random forest using all possible combinations of training and testing lidar data sets.

Overall, we find that the random forest provides accurate results, with RMSE always equal or lower than $1.51\,\mathrm{m\,s^{-1}}$. Also, we see that the model generalizes well when comparing the RMSE obtained under a round-robin scenario to the RMSE values found when using the same site for training and testing; on average, we find a 13 % increase in RMSE compared to the same-

260 site scenario. Notably, for the two NYSERDA lidars, which have the longest period of record, we find very little degradation in performance when the random forest is trained at one lidar and then tested at the other one, which is about 75 km away. To better visualize the good performance of the extrapolation model, Fig. 4 shows an example of a scatter plot of observed and machine-learning-predicted hub-height winds when the random forest is trained at NYSERDA E06 South lidar and applied at Atlantic Shores 06. The quantiles in the plot help visualize how the extrapolation model slightly overpredicts 140 m wind

265 speed for low wind speeds, and slightly underpredicts it for high wind speeds.

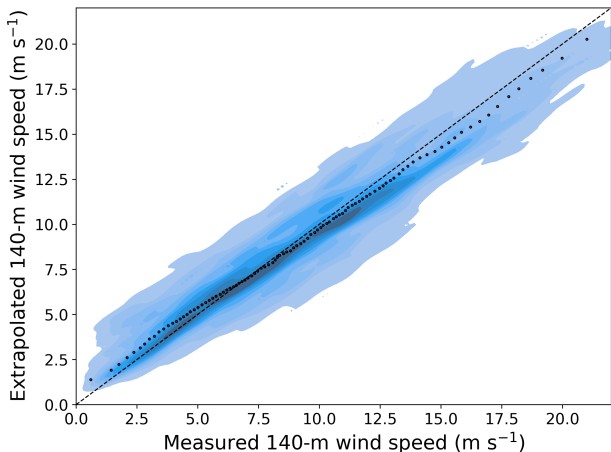

**Figure 4.** Scatter plot of observed and machine-learning-predicted 140 m hourly average wind speed at the Atlantic Shores 06 lidar when the learning algorithm is trained at the NYSERDA E06 South lidar. The color shades show density of the data, with darker colors indicating regions with more data. The black dots compare quantiles of the two samples.

When interpreting these results, it is important to also consider the correlation existing between the considered data sets. In fact, the $R^2$ coefficient between observed hourly average 140 m wind speed from the closest pairs of lidars is quite high. For example, $R^2 = 0.84$ when considering the NYSERDA E05 and E06 lidars (about 75 km from each other), and $R^2 = 0.88$ between the NYSERDA E06 and Atlantic Shores 06 lidars (about 55 km from each other). These values drop significantly when considering larger distances, so that we have $R^2 = 0.55$ between the observations of the NYSERDA E05 and Atlantic Shores 04 lidars, which are over 145 km apart. Finally, it is important to remember that the Atlantic Shores 04 and 06 lidars do not have any overlapping time in their periods of record, so that their time series can be considered independent from each other. Given these considerations, it is reasonable to expect that the existing correlations between the data sets have an impact on the good generalization skills found here, but only up to a certain level. For example, the remarkably strong generalization skill found between the two NYSERDA lidars is likely connected to a combination of their long period of records and strong autocorrelation. On the other hand, the random forest still performs well when trained and tested at lidars over 140 km apart (the NYSERDA E05 and Atlantic Shores 04 lidars), and even when trained and tested at two lidars (Atlantic Shores 04 and 06) with no overlapping period of records. Therefore, while numbers in Figure 3 are not immune from existing correlations, the overall good generalization performance of the extrapolation algorithm in the relatively limited geographical region considered in our analysis is confirmed.

The application of the random forest model also allows for a quantification of the relative importance of the various input variables used to feed the model. Table 7 shows the feature importance at the Atlantic Shores 06 lidar site. With no surprise, we find that wind speed close to the surface is the most influential variable, followed by the difference between air temperature and sea surface temperature, which is a proxy for atmospheric stability. Similar results are observed at the other lidar sites (not shown). We note that a proper feature importance quantification would require all input variables to be uncorrelated, which

**Table 7.** Predictor importance for the random forest used to extrapolate winds at 140 m above sea level at lidar Atlantic Shores 06.

| Predictor | Relative importance |
|---|---|
| Near-surface wind speed | 75.7% |
| Near-surface air temperature | 0.6% |
| Sine of near-surface wind direction | 0.7% |
| Cosine of near-surface wind direction | 1.4% |
| Sine of time of day | 1.2% |
| Cosine of time of day | 0.3% |
| Sine of month | 0.2% |
| Cosine of month | 0.2% |
| Sea surface temperature (SST) | 0.6% |
| Difference between near-surface air temperature and SST | 19.1% |

is not the case in our analysis. Therefore, the results should be considered as qualitative, and interpreted given the correlation existing between some of the input variables. For example, if the difference between air temperature and SST was not included as input, it is reasonable to expect that the relative importance of air temperature and SST would increase.

The fact that only two of the variables used as inputs to the random forest have a relatively large importance introduces the question of whether a simpler algorithm could be used to vertically extrapolate wind speeds. We test this aspect by considering two additional algorithm setups (for simplicity, we only consider the same-site approach), and summarize our results in Table 8:

- First, we test whether a comparable model accuracy can be achieved when considering a random forest that uses only the two most important variables (near-surface wind speed and difference between air temperature and SST) as inputs. We use the same range of hyperparameter and cross-validation setup used in the main analysis. We find that while the model does not overfit the data in a significant way, it has lower skills, with testing RMSE values between 0.15 - 0.40 $\mathrm{m\,s^{-1}}$ larger than what found with the original random forest setup (Table 6).

- Next, we test whether using a simpler regression algorithm with the whole set of input variables considered in the original random forest can lead to a comparable skill. We consider a multivariate linear regression, and use a ridge algorithm (Hoerl and Kennard, 1970) (`Ridge` in Python's library Scikit-learn) to constrain the multivariate regression. We use the same cross-validation setup used in the main analysis, with the only hyperparameter ($\alpha$) for the algorithm sampled between 0.1 and 10. We find that the model does not overfit the data, but it does provide significantly worse extrapolated winds, with RMSE values over 0.5 $\mathrm{m\,s^{-1}}$ larger than what found with the original random forest setup.

Therefore, we conclude that the random forest model considered in the main analysis is an appropriate choice given the complexity of the task of wind speed vertical extrapolation, despite the limited number of variables showing large values of relative importance (likely due to some correlation effects, as described above). Finally, we note that several constraints have

been applied to the complexity of the random forest used in the main analysis, in terms of the hyperparameters listed in Table 4, so that the training of the model can be easily completed on a personal computer and only takes a few minutes.

**Table 8.** Comparison of training and testing RMSE (in modeling hourly average wind speed at 140 m) at the four lidar data sets when using a random forest with a reduced number of input variables and a multivariate linear regression with the whole set of input variables.

| Lidar data set | Random forest with two inputs | | Multivariate linear regression | |
|---|---|---|---|---|
| | Training RMSE $(\mathrm{m\,s^{-1}})$ | Testing RMSE $(\mathrm{m\,s^{-1}})$ | Training RMSE $(\mathrm{m\,s^{-1}})$ | Testing RMSE $(\mathrm{m\,s^{-1}})$ |
| NYSERDA E05 | 1.32 | 1.29 | 1.75 | 1.77 |
| NYSERDA E06 | 1.32 | 1.40 | 1.87 | 1.80 |
| Atlantic Shores 04 | 1.47 | 1.55 | 1.77 | 1.85 |
| Atlantic Shores 06 | 1.55 | 1.72 | 2.08 | 1.88 |

## 6  Modeled long-term wind resource uncertainty quantification

After properly validating and assessing the generalization skills of the machine-learning-based vertical extrapolation model by leveraging the short-term lidar data, we can now apply it to extrapolate the long-term observations collected by the NDBC buoys. To do so, we train a random forest using all the lidar data sets combined to optimize the amount of training data for the model (the hyperparameters selected for this final model are listed in the leftmost column in Table 4), and then apply the trained model at each buoy location. We then compare the long-term extrapolated winds against the WRF-modeled data at 140

m above sea level (results at one buoy in Fig. 5) at each NDBC buoy location.

We finally compute the modeled wind speed uncertainty, following the steps detailed in Sect. 3.2. Table 9 shows bias and uncertainty values calculated as mean and standard deviation of the (up to) 21-year time series of residuals between modeled and extrapolated 140 m wind speed at each NDBC buoy location. We find very small biases (always smaller than $0.4\ \mathrm{m\,s^{-1}}$ in either direction) across all buoy locations. Therefore, the uncertainty in the modeled wind speed can be quantified from the

single-site WRF uncertainty values $\sigma_{\mathrm{WRF}}$ shown in the table. To these numbers, we add in quadrature a quantification of the uncertainty in the observations ($\sigma_{\mathrm{obs}}$) and of the machine learning model used to vertically extrapolate the buoy data ($\sigma_{\mathrm{ML}}$). Once again, following the lidar uncertainty assessment in OceanTech Services/DNV GL (2020), we consider $\sigma_{\mathrm{obs}} = 0.29\ \mathrm{m\,s^{-1}}$ (which is slightly different from what was used in Sect. 4 because this time we are calculating the mean wind speed over the full period of record of all lidar data sets). We quantify the extrapolation model uncertainty in terms of the mean RMSE obtained

under all the site combinations considered in the round-robin validation (i.e., the mean of all the off-diagonal values in the matrix in Fig. 3) so that $\sigma_{\mathrm{ML}} = 1.38\ \mathrm{m\,s^{-1}}$.

We find that at all but one buoy, the total uncertainty in modeled 140 m wind speed is equal to or slightly lower than 3 $\mathrm{m\,s^{-1}}$ (roughly equal to 30% of the mean 140-m wind speed, see Table 2). The uncertainty increases as the distance from the lidars, used to train the machine learning model, increases. As discussed in Sect. 3.2, this aspect is directly visible in the $\sigma_{\mathrm{WRF}}$

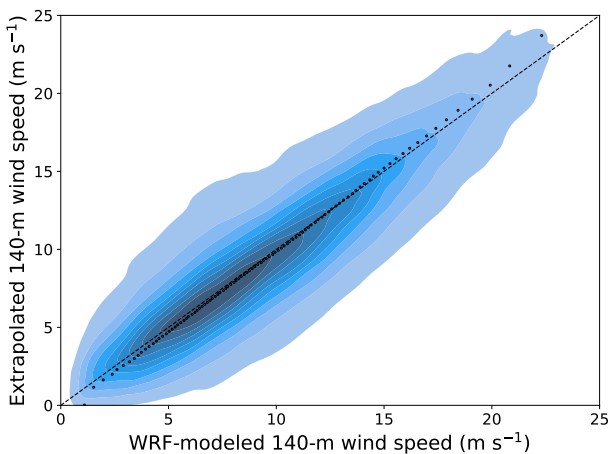

**Figure 5.** Scatter plot of 21-year WRF-modeled and machine-learning-predicted 140 m hourly average wind speed at the location of the 44025 NDBC buoy. The color shades show density of the data, with darker colors indicating regions with more data. The black dots compare quantiles of the two samples.

**Table 9.** Twenty-year model bias and model uncertainty in 140 m wind speed at the location of the NDBC buoys considered in this study.

| NDBC buoy | Bias ($\mathrm{m\,s^{-1}}$) | WRF uncertainty ($\mathrm{m\,s^{-1}}$) | Total uncertainty ($\mathrm{m\,s^{-1}}$) |
|---|---|---|---|
| 44008 | 0.38 | 2.66 | 3.00 |
| 44009 | 0.12 | 2.60 | 2.96 |
| 44017 | 0.15 | 2.50 | 2.87 |
| 44018 | 0.22 | 3.59 | 3.86 |
| 44020 | 0.29 | 2.65 | 2.99 |
| 44025 | -0.06 | 2.52 | 2.89 |
| 44065 | 0.17 | 2.63 | 2.99 |
| 44066 | 0.11 | 2.46 | 2.84 |

uncertainty component. Specifically, buoy 44018 has the largest uncertainty, which is consistent with this buoy being separated from all the lidars by Cape Cod; it is reasonable to expect that the atmospheric conditions at this buoy site are considerably different from what was used to train the machine learning model. Also, we note how the total uncertainty values obtained here are about $1~\mathrm{m\,s^{-1}}$ higher than what was found from the short-term direct comparison between lidar observations and WRF modeled data in Sect. 4. While the impact of different lengths of analysis cannot be ruled out, this comparison shows
how having access to the long-term lidar observations would be extremely beneficial in allowing a more direct quantification (leading to lower values) of the model uncertainty for long-term wind resource assessment purposes.

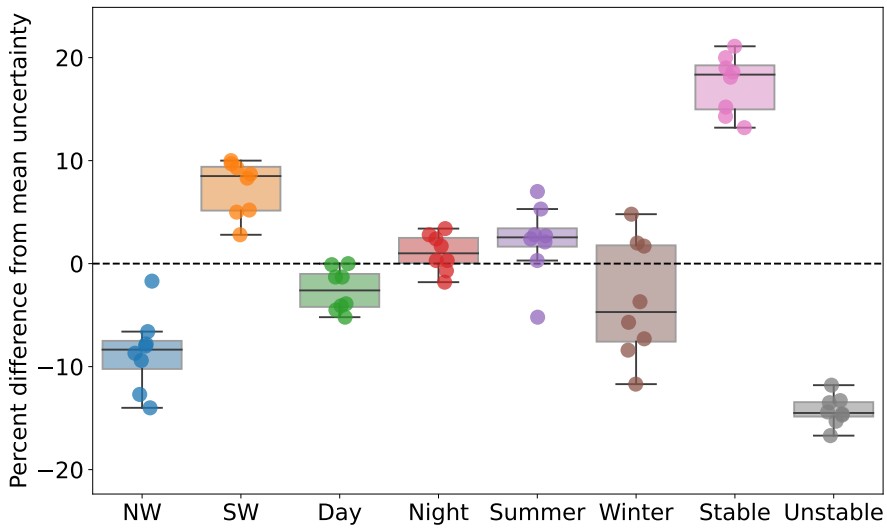

**Figure 6.** Box plot showing how the modeled 140 m wind speed uncertainty varies as a function of wind direction, time of day, season, and atmospheric stability conditions. For each buoy location, results are expressed as percent difference from the mean uncertainty values (rightmost column in Table 9).

Finally, we focus on the variability of the quantified uncertainty and segregate results by time of day (9 a.m.–4 p.m. local time vs. 9 p.m.–4 a.m. local time), season (June, July, August vs. December, January, February), wind direction (180°–270° vs. 270°–360°, which are the two dominant wind direction regimes in the region (Pronk et al., 2022)), and atmospheric stability conditions (quantified in terms of the modeled inverse Obukhov length $L^{-1}$ at 2 m above sea level, where we simply consider stable conditions for $L^{-1} > 0$ m$^{-1}$ and unstable conditions for $L^{-1} < 0$ m$^{-1}$). We summarize our results in the box plots in Fig. 6. The largest difference in modeled wind speed uncertainty is for stable conditions, which are generally more challenging to model compared to unstable conditions. Pronk et al. (2022) showed that stable conditions in this region are dominant in the summer, and Bodini et al. (2019) showed that southwesterly winds are dominant in the summer months. In fact, we find a larger wind speed uncertainty for southwesterly winds and in the summer (although winter shows a significant scatter among the buoys). Finally, nighttime uncertainty is larger than daytime, although the difference between the two is limited.

## 7 Conclusions

The National Renewable Energy Laboratory has released a state-of-the-art 21-year wind resource assessment product for all the offshore regions in the United States. Because of its numerical nature, this data set has inherent uncertainty, the quantification of which is of primary importance for stakeholders aiming to use this data set to contribute to offshore wind energy growth. In our analysis, we have shown the limits of quantifying model uncertainty in terms of the variability of a model ensemble, which in our case captured only roughly half of the total model uncertainty. Instead, we recommend leveraging observations

to fully capture NWP model uncertainty. In the absence of long-term observed wind speeds at hub height, we have proposed a methodological pipeline to vertically extrapolate near-surface winds from long-term buoy observations using machine learning.

We adopt a random forest model, using a number of atmospheric variables measured near the surface as inputs to the regression algorithm. Our approach was well validated across the mid-Atlantic region, and we showed that using a significantly simpler model (either in terms of the regression algorithm itself, or the number of input variables used) would significantly reduce the accuracy of the extrapolated winds. The total model uncertainty we observed in hub-height hourly wind speed was, on average, just below $3 \mathrm{~m~s^{-1}}$ (about 30% of the mean observed winds). This number is not negligible, especially considering that wind

turbine power production is roughly related to the cube of wind speed, but several opportunities exist to reduce this uncertainty in the future.

This analysis is one of many examples of the synergy between NWP models and observations, which points to the multiple interconnections between the two. A larger number of long-term observations are needed to both quantify and, in the long term, reduce the inherent uncertainty of numerical models. In fact, we observe that the uncertainty in the modeled data increases as we

move away from the observational data sets used to train the machine learning algorithm. Having a larger number of sites with available hub-height observations covering a variety of atmospheric conditions would allow for the machine learning model to more accurately represent hub height conditions across a wider region. In this context, the sharing of additional proprietary observational data sets should be considered, and the long-term advantages resulting from better numerical modeling should be kept in mind when assessing the overall balance between costs and benefits of such data-sharing initiatives. In the future,

the choice of the learning algorithm as well as of the input variables can be explored in more detail, for example by testing a larger number of regression models than what considered here. Also, a similar analysis can be performed for other offshore regions where both a long-term numerical wind resource assessment product and enough observations to assess uncertainty are available.

*Data availability.* NREL's long-term wind resource data sets can be found at https://doi.org/10.25984/1821404. The WRF namelist is stored

at https://doi.org/10.5281/zenodo.7814365. NDBC buoy observations can be downloaded from https://www.ndbc.noaa.gov. Observations from the NYSERDA floating lidars can be accessed at https://oswbuoysny.resourcepanorama.dnv.com. Atlantic Shores lidar observations can be downloaded from https://erddap.maracoos.org/erddap/tabledap.

*Author contributions.* Nicola Bodini: conceptualization, methodology, formal analysis, writing (original draft), visualization, supervision, project administration. Simon Castagneri: formal analysis, writing (review), and editing. Mike Optis: conceptualization, funding acquisition.

*Competing interests.* The authors declare they have no competing interests.

*Acknowledgements.* The authors would like to thank the members of the NOWRDC project advisory board, and in particular Nicolai Nygaard, for the constructive feedback that helped shape the analysis. A portion of this research was performed using computational resources sponsored by the U.S. Department of Energy's Office of Energy Efficiency and Renewable Energy and located at the National Renewable Energy Laboratory.

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
