# Peer review of "Long-term uncertainty quantification in WRF-modeled offshore wind resource off the US Atlantic coast"

_Wind Energy Science, 2023_

## Referee Comment (RC1)

**Referee comment to**

**Title: Long-term uncertainty quantification in WRF-modeled offshore wind resource off the US Atlantic coast**
Author(s): Nicola Bodini and Simon Castagneri

**General comments:**

Overall, the manuscript is well written and structured in an understandable way. The introduction provides enough background and references to connect the article to the current state-of-the-art and the methodology section provides a high degree of details to understand and follow the workflow. Some elements in the methodology and data descriptions however require, in my opinion, additional description or additional references. While the analysis in general is presented concise and in a convincing way, I see the need for further investigation & provision of statistics with respect to the claimed generalization of the machine learning approach (see specific comment #2). Some additional words on limitations of the methodologys range of application would be appreciated as well. While I would suggest also some minor changes in the text and data presentation, my recommendation is publication after the questions/comments below have been addressed adequately.

**Specific comments:**

general remarks:

**1 Usage of the term "boundary condition and parametric uncertainty"**

The authors introduce above mentioned term to describe the share of uncertainty that can described by NWP ensemble runs appearing several times in the manuscript. This term, however, can be highly misleading and misinterpreted especially in the context of regional NWP where the term "boundary conditions" and boundary condition "uncertainties" are used in a different context. I would suggests a term like "Ensemble-derived uncertainty" or something along those lines to avoid misinterpretation.

**2 Generalization of the machine learning approach**

Section 5 describes the details of the validation of the machine learning approach. While the round-robin cross-validation approach shows promising behavior, the authors acknowledge the impact of spatial correlation due to the close vicinity of locations of validation. Here, I think it very crucial to quantify this impact by e.g. calculating mutual correlation coefficients of the Lidar time series to fully understand how independent the training and validation data actually are (maybe presented in a similar manner to Fig.4). This would also provide more details to the generalization skill and to what extent the high generalization skill is achieved purely due to high correlation of training and validation data. In this context, it would be also valuable if the mutual distance between the Lidars could be stated.

**3 More details in description of numerical WRF setup**

While the setup is described fairly detailed, the following information is missing:
- Nudging is mentioned in l. 72, but it remains unspecified if grid or spectral nudging has been used. Please specify and provide details on parameter settings if they differ from the default settings in WRF.
- Land surface model, Microphysics, Longwave/Shortwave radiation, topographic data base and land use data in Table 1 lack references. Please add them for completeness in line with the other specifications in Table 1.

**4 Limitations of the proposed methodology**

In the current version of the manuscript, very little is talked about the limitations of this methodology. While there is a statement made about the atmospheric conditions at the buoy north of Cape Cod (l. 244), a more in-dept critical elaboration is required in my opinion (maybe as additional paragraph in Sect. 6). This concerns especially the reliability / trustworthiness of the method when the random forest is applied to locations that are very different from the training data (geographic location, distance to training data, atmospheric conditions).

remarks addressing specific lines or sections:

l. 8: Since it is the abstract, please be specific about your used method (random forest) instead of the generic term "machine learning technique".

l. 22: The stated reference for the currently installed offshore wind farms in the US is around 7 years old, please update with newer reference (maybe GEWCs Global Wind Report) to confirm and to use more up-to-date data.

l. 31: The reference (Skamarock et al. 2008) points towards Version 3 of the WRF model, but your WRF version seems to be 4.2.1 (Table 1). Is there a particular reason why the Version 3 reference is used here and not Version 4? Otherwise, please update.

l. 125/126: I would recommend to mention here again that the variables used for training are coming from the Lidar to avoid any ambiguity about the input for the training process.

l. 135 (footnote): The part " […] which are both needed because each value of sine only (or cosine only) is linked to two different values of the cyclical variable" is unclear to me. What does it mean? Please consider elaboration or reformulation.

l. 149: What exactly do you mean by "typical single-site uncertainty"? Is this the averaged standard deviation of the residual time series for a particular location or something else? Please elaborate.

l. 270: Replace "wind energy" with "wind turbine power production"

**Technical corrections:**

Language corrections:

l. 3 […] heavier relative weight [...] → […] heavier is the relative weight […]

l. 144 Then, to assess the uncertainty → To assess the uncertainty

l. 161 Introduction → introduction

l. 256 […] to numerically model […] → […] to model [...]

Figures:

Fig. 1: For completeness, please state in the figure caption which markers indicate Lidar locations and which markers indicate buoy locations. This is currently unclear by looking at the figure only.

Fig. 6: I would suggest to transform this figure to a table since the bars for the parameters in the middle do not convey much information. For completeness, I would also suggest adding the explanation of "SST" to the caption similar to "WS" and "WD".

---

## Referee Comment (RC2)

**Long-term uncertainty quantification in WRF-modeled offshore wind resource off the US Atlantic coast**
by
**N. Bodini and S. Castagneri**

The paper nicely brings together methods and simulations from previously peer-reviewed papers by the first author to answer the question of uncertainty quantification connected to offshore long-term modelled hub-height wind speed in the case of limited (only near surface) observational data. The method considers modelling uncertainty as well as observationally based uncertainties connected to near surface observations uncertainties and height extrapolation uncertainties using a peer-reviewed machine learning approach.

**General comment**

I find the paper interesting and providing new and original material. It is clearly within the scope of WES and introduces a novel approach to uncertainty quantification. By backtracking references to previous papers, the methods and simulations are in principle repeatable. However, some more information on the machine learning algorithm such as choice of hyperparameters etc. would be beneficial for better reproducibility.

The paper is written in a clear and well-structured way. It would, however, benefit from a clearer statement of objectives in the final part of the introduction.
As errors are depending on averages in time and space it should also be more clearly emphasized in both the abstract, introduction and conclusions that results are for hourly average windspeeds.

A final general remark is that most results are given in terms of rmse (m/s) and for readers unfamiliar with the climatology of the region in is difficult to assess if uncertainties in the order of 2-3 m/s are large or small. Some indication of relative uncertainties or typical hour-to-hour wind speed variability would help alleviate this.

**Specific comments**

- Abstract: In the abstract there is only one sentence on the results. A slightly more elaborate description of the results (such as how errors are connected to stability etc.) would be beneficial.

- Introduction: The introduction provides a thorough review of uncertainties in NWP modelling, but relatively little is given on the observational uncertainties ranging from instrument uncertainties to representation uncertainty. A possible starting point could be the below review of forecasting errors.
  https://www.sciencedirect.com/science/article/pii/S1364032122004221

- Introduction: The final part of the introduction would benefit from a clearer statement of objectives and sub-objectives to make it clear for the reader what the objectives are.

- Section 2.1: The paper references to a previous paper for details about the numerical data. This is fine, but I would have preferred that information such as horizontal resolution, number of levels and output frequency is given in the text and not only stated in the table.

- Section 2.2.1: The authors list the different lidar data used and state that "proprietary quality checks" are done. A few sentences or a reference to what the proprietary quality check contains would be useful.

- Section 2.2.1: I found it unclear what was done in terms of averaging for the lidar data. The authors state in the section 2.2.1 that "*We kept only hourly time stamps where 140 m wind speed, near-surface wind speed, near-surface wind direction, air temperature, and sea surface temperature were all available.*" Does this mean that when relations between near surface and 140m winds are estimated based on 5 or 10 min. values, or are there some kind of hourly averaging done before the relations are estimated? If the later is the case please not how missing data is treated in the averaging.

- Section 2.2.2: How is missing data treated in the averaging  the NDBC buoy observations?

- Section 3.1: This describes the Machine learning algorithm for wind speed vertical extrapolation. The input are hourly averaged NDBC buoy observations, but it is unclear if the lidar data used for learning is 10 min or hourly averaged lidar data?

- Section 3.1: The authors perform a 5-fold cross validation using a consecutive 20% of the observations. The test data should be unconnected to the training data for a realistic estimation of a possible training to testing degradation of the quality. As windspeeds may easily have lag correlations, the authors should justify that the test data indeed is unconnected to the training data. Any remaining relations between the training and test data could seriously hamper the error statistics. A possible test would be to see if splitting the data chronologically (i.e. 20% of the years used for testing) gives a similar result as the selected validation method.
- The authors should also state both the training and testing validation in order to discuss possible overfitting issues.

- Section 3.1: The authors test different hyperparameters, but the final choices of parameters are not stated.

- Section 3.2: In the total uncertainty estimates the uncertainty components are added in quadrature (equation 1). This way of estimating the total errors is only valid if the

errors are not correlated. Please add an analysis showing that the error components are uncorrelated to justify equation 1.

- Section 4: The authors state that "*This comparison clearly confirms how an NWP model's boundary condition and parametric uncertainty, which can be quantified from the variability across a numerical ensemble, can only quantify a limited component of the full model uncertainty*", but how is the ensemble uncertainty quantified? The 16 member ensemble will be highly correlated and this have to be reflected in the ensemble uncertainty estimation through the error covariance.

- Section 5: This consist of the Machine learning wind speed vertical extrapolation validation. A key question that pops up in the round-robin validation is how correlated the sites are. The results It is not easy to understand if the slight increase in rmse of around 12.5% for the round-robin validation compared to the same-site validation is high or low, as long as there is no analysis of among-site correlation.

- Section 5: Figure 6 gives the quantification of the relative importance of the various input variables. It is not stated, but I guess this is the random forest out-of-bag estimate? If so, this analysis requires that the features are uncorrelated, so this has to be established before any clear conclusions can be drawn from the analysis. If they are correlated, this should be stated with the possible influences this may have on the results.

- Section 5: The relative importance analysis (fig 6) indicates that only two features account for most of the predictability. If this is the case, one could argue that the machine learning model, could be heavily simplified. This could be investigated, by looking at the quality degradation between a model using all features and one using only the two main features. Using for example AIC (Aike Information Criterion) or other measures that penalizes unnecessary complicated models, one could get an idea if having all features is worth it compared to a simpler model.

- Section 5: It is unclear what the benefit of the random forest model is compared to a simple more transparent multilinear regression model? Is the added complexity worth it? It would be very interesting if the authors could provide an estimate of the added value of the non-linear random forest model compared to a simple multilinear regression model with only two features.

- Section 6: The author states that uncertainty increases as the distance from the lidars, used to train the machine learning model, increases. But this is not incorporated in the extrapolation uncertainty. Instead it may turn up as part of the WRF uncertainty as the model is compared to data which have gradually lower quality as the validation is moving away from the lidar used in extrapolation training. This questions the attribution of uncertainty to the different terms (wrf vs total). A discussion on this would help the readers to reflect on how the different uncertainty terms should be interpreted and the fact that they might not be independent from each other.

- The conclusion would benefit from some critical discussion on the choice of methods, possible limitations, and outstanding research questions.

**Figures**
- Figure 1: State the difference between diamonds and dots in the caption.

- Figure 3: Consider skipping this.

- Figure 5 and 7: The scatter plot caption needs a sentence about the color coding indicating density of the data. State that the data shown is hourly averaged wind speeds.

- Figure 5 and 7: Plotting some quantile-quantile values as dots on top of the current plot would make it easier to see any systematic differences in the quantiles.

---

## Author Comment (AC1)

*In this document, the reviewer's comments are in black, the authors' responses are in red.*

**General comments:**

Overall, the manuscript is well written and structured in an understandable way. The introduction provides enough background and references to connect the article to the current state-of-the-art and the methodology section provides a high degree of details to understand and follow the workflow. Some elements in the methodology and data descriptions however require, in my opinion, additional description or additional references. While the analysis in general is presented concise and in a convincing way, I see the need for further investigation & provision of statistics with respect to the claimed generalization of the machine learning approach (see specific comment #2). Some additional words on limitations of the methodologys range of application would be appreciated as well. While I would suggest also some minor changes in the text and data presentation, my recommendation is publication after the questions/comments below have been addressed adequately.

We thank the reviewer for their thoughtful comments, which we addressed in our response to the specific comments below.

**Specific comments:**

general remarks:

**1 Usage of the term "boundary condition and parametric uncertainty"**
The authors introduce above mentioned term to describe the share of uncertainty that can described by NWP ensemble runs appearing several times in the manuscript. This term, however, can be highly misleading and misinterpreted especially in the context of regional NWP where the term "boundary conditions" and boundary condition "uncertainties" are used in a different context. I would suggests a term like "Ensemble-derived uncertainty" or something along those lines to avoid misinterpretation.
We now use "ensemble-derived uncertainty" as suggested throughout the paper. We kept only one instance of the old terminology to specify that this was the terminology used in the Bodini et al. 2021 article.

**2 Generalization of the machine learning approach**
Section 5 describes the details of the validation of the machine learning approach. While the round-robin cross- validation approach shows promising behavior, the authors acknowledge the impact of spatial correlation due to the close vicinity of locations of validation. Here, I think it very crucial to quantify this impact by e.g. calculating mutual correlation coefficients of the Lidar time series to fully understand how independent the training and validation data actually are (maybe presented in a similar manner to Fig.4). This would also provide more details to the generalization skill and to what extent the high generalization skill is achieved purely due to high correlation of training and validation data. In this context, it would be also valuable if the mutual distance between the Lidars could be stated.
We added the following paragraph to Section 5:

When interpreting these results, it is important to also consider the correlation existing between the considered data sets. In fact, the $R^2$ coefficient between observed hourly average 140 m wind speed from the closest pairs of lidars is quite high. For example, $R^2 = 0.84$ when considering the NYSERDA E05 and E06 lidars (about 75 km from each other), and $R^2 = 0.88$ between the NYSERDA E06 and Atlantic Shores 06 lidars (about 55 km from each other). These values drop significantly when considering larger distances, so that we have $R^2 = 0.55$ between the observations of the NYSERDA E05 and Atlantic Shores 04 lidars, which are over 145 km apart. Finally, it is important to remember that the Atlantic Shores 04 and 06 lidars do not have any overlapping time in their periods of record, so that their time series can be considered independent from each other. Given these considerations, it is reasonable to expect that the existing correlations between the data sets have an impact on the good generalization skills found here, but only up to a certain level. For example, the remarkably strong generalization skill found between the two NYSERDA lidars is likely connected to a combination of their long period of records and strong autocorrelation. On the other hand, the random forest still performs well when trained and tested at lidars over 140 km apart (the NYSERDA E05 and Atlantic Shores 04 lidars), and even when trained and tested at two lidars (Atlantic Shores 04 and 06) with no overlapping period of records. Therefore, while numbers in Figure 3 are not immune from existing correlations, the overall good generalization performance of the extrapolation algorithm in the relatively limited geographical region considered in our analysis is confirmed.

Finally, we added information on the distance between the lidars in the caption of Figure 1: "The distance between the two NYSERDA lidars is about 75 km, the two Atlantic Shores lidars are about 20 km from each other, and finally the distance between the NYSERDA E05 and Atlantic Shores 04 lidars is about 145 km."

**3 More details in description of numerical WRF setup**
While the setup is described fairly detailed, the following information is missing:
- Nudging is mentioned in l. 72, but it remains unspecified if grid or spectral nudging has been used. Please specify and provide details on parameter settings if they differ from the default settings in WRF.
- Land surface model, Microphysics, Longwave/Shortwave radiation, topographic data base and land use data in Table 1 lack references. Please add them for completeness in line with the other specifications in Table 1.
We added a specification that we used spectral nudging, and references to all missing entries in the table.

**4 Limitations of the proposed methodology**
In the current version of the manuscript, very little is talked about the limitations of this methodology. While there is a statement made about the atmospheric conditions at the buoy north of Cape Cod (l. 244), a more in-dept critical elaboration is required in my opinion (maybe as additional paragraph in Sect. 6). This concerns especially the reliability / trustworthiness of the method when the random forest is applied to locations that are very different from the training data (geographic location, distance to training data, atmospheric conditions).
We have added a whole new paragraph to critically address the choice of the random forest model, and whether using fewer input variables could be enough to provide an accurate uncertainty quantification, to more critically explore some of the potential limitations of the proposed methodology.
We also added some text to the conclusions to further stress that the results are location-dependent, and how having access to more and more hub-height observations would be beneficial in lowering the uncertainty associated to the machine learning extrapolation model.

Remarks addressing specific lines or sections:

l. 8: Since it is the abstract, please be specific about your used method (random forest) instead of the generic term "machine learning technique".

Changed as suggested.

l. 22: The stated reference for the currently installed offshore wind farms in the US is around 7 years old, please update with newer reference (maybe GEWCs Global Wind Report) to confirm and to use more up-to-date data.
We changed the sentence to "While the United States currently only have 42 MW of installed offshore wind capacity (Global Wind Energy Council, 2023), …".

l. 31: The reference (Skamarock et al. 2008) points towards Version 3 of the WRF model, but your WRF version seems to be 4.2.1 (Table 1). Is there a particular reason why the Version 3 reference is used here and not Version 4? Otherwise, please update.
Thanks for catching this – we updated the reference.

l. 125/126: I would recommend to mention here again that the variables used for training are coming from the Lidar to avoid any ambiguity about the input for the training process.
We added the following specification: "We use the following observed variables as inputs to the model" (we did not specify "lidar" as some variables are technically coming from other instruments mounted on the lidar buoys).

l. 135 (footnote): The part " [...] which are both needed because each value of sine only (or cosine only) is linked to two different values of the cyclical variable" is unclear to me. What does it mean? Please consider elaboration or reformulation.
We added details and rephrased this as "To preserve the cyclical nature of this variable, we calculate and include as inputs its sine and cosine. We note that both sine and cosine are needed to identify a specific value of the cyclical variable, because each value of sine only (or cosine only) is linked to two different values of the cyclical variable. For example, the sine of wind direction is 0 for both 90∘and 270∘, but once their (different) cosines are introduced as well, the two can be identified in a univocal way".

l. 149: What exactly do you mean by "typical single-site uncertainty"? Is this the averaged standard deviation of the residual time series for a particular location or something else? Please elaborate.
We added the following specification "(i.e., the average of each site's standard deviation of the residuals)".

l. 270: Replace "wind energy" with "wind turbine power production"
Changed.

**Technical corrections:**
Language corrections:
l. 3 [...] heavier relative weight [...] → [...] heavier is the relative weight [...] Rephrased as "and the resulting heavier relative weight".
l. 144 Then, to assess the uncertainty → To assess the uncertainty  Changed.
l. 161 Introduction → introduction Changed.
l. 256 [...] to numerically model [...] → [...] to model [...] Changed.

Figures:
Fig. 1: For completeness, please state in the figure caption which markers indicate Lidar locations and which markers indicate buoy locations. This is currently unclear by looking at the figure only.
We added the following sentence to the caption "Lidar locations are shown as diamonds, NDBC buoys are shown as dots."

Fig. 6: I would suggest to transform this figure to a table since the bars for the parameters in the middle do not convey much information. For completeness, I would also suggest adding the explanation of "SST" to the caption similar to "WS" and "WD".
Changed.

---

## Author Comment (AC2)

*In this document, the reviewer's comments are in black, the authors' responses are in red.*

We thank the reviewer for their thoughtful comments.

The paper nicely brings together methods and simulations from previously peer-reviewed papers by the first author to answer the question of uncertainty quantification connected to offshore long-term modelled hub-height wind speed in the case of limited (only near surface) observational data. The method considers modelling uncertainty as well as observationally based uncertainties connected to near surface observations uncertainties and height extrapolation uncertainties using a peer-reviewed machine learning approach.

**General comment**

I find the paper interesting and providing new and original material. It is clearly within the scope of WES and introduces a novel approach to uncertainty quantification. By backtracking references to previous papers, the methods and simulations are in principle repeatable. However, some more information on the machine learning algorithm such as choice of hyperparameters etc. would be beneficial for better reproducibility.
The description of the machine learning algorithm now includes detailed information about the hyperparameters (new table added and relevant text in the main section):

**Table 4.** Algorithm hyperparameters considered for the random forest, their sampled values in the cross validation, and chosen value in the final version of the model used in Sect. 6.

| Hyperparameter | Sampled Values | Selected value |
|---|---|---|
| Number of estimators | 10–20 | 16 |
| Maximum depth | 1–10 | 9 |
| Maximum number of features | 1–10 | 8 |
| Minimum number of samples to split | 2–11 | 10 |
| Minimum number of samples for a leaf | 1–15 | 6 |

The paper is written in a clear and well-structured way. It would, however, benefit from a clearer statement of objectives in the final part of the introduction.
As errors are depending on averages in time and space it should also be more clearly emphasized in both the abstract, introduction and conclusions that results are for hourly average windspeeds.
We now specify in abstract, introduction and conclusions that results are for hourly wind speeds. We also expanded the final part of the introduction as:

In our analysis, we present and validate a novel methodology to assess long-term (in our case, 21-year) uncertainty quantification for modeled wind speed in the mid-Atlantic region of the United States by leveraging available observations of offshore wind. While in our analysis we focus on the U.S. mid-Atlantic domain, the methodology could be applied in other offshore regions as well. In Sect. 2 we describe our long-term WRF simulations, and the lidar and buoy observations that we leverage to assess uncertainty in the modeled data set. In Sect. 3 we present our proposed methodology to assess long-term uncertainty in modeled wind speed by comparing it with vertically extrapolated observed winds. In Sect. 4 we dive deeper into the already mentioned topic of using numerical ensembles to quantify uncertainty and provide a demonstration of the limits of such an approach. We accurately validate our uncertainty quantification approach in Sect. 5, present the main results of our long-term uncertainty quantification (in terms of uncertainty in hourly average wind speed) in Sect. 6, and conclude our analysis in Sect. 7.

A final general remark is that most results are given in terms of rmse (m/s) and for readers unfamiliar with the climatology of the region in is difficult to assess if uncertainties in the order of 2-3 m/s are large or small. Some indication of relative uncertainties or typical hour-to-hour wind speed variability would help alleviate this.

Good point – we now included the following table, and added sentences here and there throughout the manuscript to remind the reader how the uncertainty values (expressed as RMSE) compare to the observed mean wind speed.

**Table 2.** Mean 140-m mean speed from the four lidar data sets (after all the quality checks described in Section 2.2.1 were applied).

| Lidar | Mean 140-m wind speed (m/s) |
|---|---|
| NYSERDA E05 North | 9.38 |
| NYSERDA E06 South | 9.99 |
| Atlantic Shores 04 | 9.44 |
| Atlantic Shores 06 | 9.69 |

**Specific comments**
- Abstract: In the abstract there is only one sentence on the results. A slightly more elaborate description of the results (such as how errors are connected to stability etc.) would be beneficial.

  We rephrased the final part of the abstract as: "We find that the mean 21-year uncertainty in 140 m hourly average wind speed is slightly lower than 3 m/s (roughly 30% of the mean observed wind speed) across the considered region. Atmospheric stability is strictly connected to the modeled wind speed uncertainty, with stable conditions associated with an uncertainty which is, on average, about 20% larger than the overall mean uncertainty."

- Introduction: The introduction provides a thorough review of uncertainties in NWP modelling, but relatively little is given on the observational uncertainties ranging from instrument uncertainties to representation uncertainty. A possible starting point could be the below review of forecasting errors.
  https://www.sciencedirect.com/science/article/pii/S1364032122004221
  We added the following paragraph:

It is important to remember that, while observational data sets are essential for a proper quantification of numerical uncertainty, they also come with their own uncertainty, which should therefore be considered in the overall numerical uncertainty quantification. Observational uncertainty stems from a variety of factors (Yan et al., 2022). The size and representativeness of the available observations is a primary factor to keep in mind, especially when less than one year of data is available for use, and the seasonal cycle of atmospheric variables cannot be accurately captured. Also, observations come with inherent instrumental uncertainty, whose general guidelines are well described in the ISO Guide to the expression of Uncertainty in Measurements (JCGM 100:2008, 2008b, a), often referred to as "GUM". Specific considerations will then apply for each specific instrument, so that for example lidars will have different uncertainties compared to traditional anemometers.

- Introduction: The final part of the introduction would benefit from a clearer statement of objectives and sub-objectives to make it clear for the reader what the objectives are.
  See our answer to the general comment above.

- Section 2.1: The paper references to a previous paper for details about the numerical data. This is fine, but I would have preferred that information such as horizontal resolution, number of levels and output frequency is given in the text and not only stated in the table.
  We added the following sentence to the main text: "The model is run from January 2000 to December 2020, at 2-km horizontal resolution, 5-minute temporal resolution, with nine vertical levels in the lowest 200 m, using the model setup illustrated in detail in Table 1". Also, while not ideal, we will publish a detail report on the WRF data set itself in summer 2023.

- Section 2.2.1: The authors list the different lidar data used and state that "proprietary quality checks" are done. A few sentences or a reference to what the proprietary quality check contains would be useful.
  We also wish we had access to more information on this. We reached out to instrument mentors, and they told us they cannot share any additional information about what was performed, unfortunately.

- Section 2.2.1: I found it unclear what was done in terms of averaging for the lidar data. The authors state in the section 2.2.1 that "*We kept only hourly time stamps where 140 m wind speed, near-surface wind speed, near-surface wind direction, air temperature, and sea surface temperature were all available.*" Does this mean that when relations between near surface and 140m winds are estimated based on 5 or 10 min. values, or are there some kind of hourly averaging done before the relations are estimated? If the later is the case please not how missing data is treated in the averaging.
  The section now reads: "For all lidars, we calculate hourly averages of all the relevant variables. If a variable is missing from some 10-minute periods, the hourly average value is still calculated using the available data within that 60-minute period. We kept only hourly time stamps where hourly average values of 140 m wind speed, near-surface wind speed, near-surface wind direction, air temperature, and sea surface temperature were all available."

- Section 2.2.2: How is missing data treated in the averaging the NDBC buoy observations?

We added the following: "As done for the lidar data, if a variable is missing from some 10-minute intervals, the hourly average value is still calculated using the available data within that 60-minute period. Only hourly time stamps where we have valid hourly average values for all the relevant variables are kept for the analysis."

- Section 3.1: This describes the Machine learning algorithm for wind speed vertical extrapolation. The input are hourly averaged NDBC buoy observations, but it is unclear if the lidar data used for learning is 10 min or hourly averaged lidar data?
  This part now reads: "We train the regression model to predict hourly average wind speed at 140 m. We use the following observed variables as inputs to the model, all as hourly averages:"

- Section 3.1: The authors perform a 5-fold cross validation using a consecutive 20% of the observations. The test data should be unconnected to the training data for a realistic estimation of a possible training to testing degradation of the quality. As windspeeds may easily have lag correlations, the authors should justify that the test data indeed is unconnected to the training data. Any remaining relations between the training and test data could seriously hamper the error statistics. A possible test would be to see if splitting the data chronologically (i.e. 20% of the years used for testing) gives a similar result as the selected validation method.
  Good point – we tested this as suggested and found no significant impact (the testing RMSE with the proposed approach was actually always a tad lower than what we found with our customized approach). We added a paragraph to this section: "The chosen splitting approach in the cross-validation ensures that short-term autocorrelation in the data does not artificially increase the measured skill of the algorithm (as it would happen if training and testing data sets were randomly chosen without imposing a consecutive data requirement). However, potential lag correlation in the data could still play a role. Therefore, we tested whether using a single, consecutive 20% of the data for testing leads to significantly different results in terms of model accuracy. We tested this on the two NYSERDA lidars (because they both span a period of record longer than one year, and therefore can still capture a full seasonal cycle in their training phase even when a single 20% of the data is kept aside for testing), and found no significant difference in the model performance.".

- The authors should also state both the training and testing validation in order to discuss possible overfitting issues.
  Thank you for raising this concern, which admittedly we did not check while performing our analysis. Indeed, we were seeing some overfitting of the data with the set of hyperparameters originally considered. We have now simplified the model (in terms of number of trees and tree depth), and we are not seeing overfitting anymore. We added the following to Section 3.1: "We note that the range of hyperparameters used limits the complexity of the random forest, and limits the computational resources needed to train the model. We tested using a larger forest with deeper trees, but that led to overfitting the available training data."
  We have also added the following discussion to Section 5:

First, we verify that the learning algorithm we have chosen does not overfit the data. To do so, we compare the training and testing RMSE at the four lidar data sets in Table 6. The fact we do not systematically see significantly larger testing RSME confirms that the random forest does not overfit the data at any of the lidar sites, so that we continue in our validation by assessing its generalization skills.

**Table 6.** Comparison of training and testing RMSE (in modeling hourly average wind speed at 140 m) at the four lidar data sets.

| Lidar data set | Training RMSE (m s$^{-1}$) | Testing RMSE (m s$^{-1}$) |
|---|---|---|
| NYSERDA E05 | 1.09 | 1.14 |
| NYSERDA E06 | 1.21 | 1.18 |
| Atlantic Shores 04 | 1.17 | 1.22 |
| Atlantic Shores 06 | 1.27 | 1.33 |

We updated all the results in the paper accordingly (no major changes).

- Section 3.1: The authors test different hyperparameters, but the final choices of parameters are not stated.
  We added a column to Table 4 (included above) showing the chosen values for the final model, and reference is given in Section 6 ("(the hyperparameters selected for this final model are listed in the leftmost column in Table 4)").

- Section 3.2: In the total uncertainty estimates the uncertainty components are added in quadrature (equation 1). This way of estimating the total errors is only valid if the errors are not correlated. Please add an analysis showing that the error components are uncorrelated to justify equation 1.
  Please see our answer to the second to last specific comment in this list.

- Section 4: The authors state that "*This comparison clearly confirms how an NWP model's boundary condition and parametric uncertainty, which can be quantified from the variability across a numerical ensemble, can only quantify a limited component of the full model uncertainty*", but how is the ensemble uncertainty quantified? The 16 member ensemble will be highly correlated and this have to be reflected in the ensemble uncertainty estimation through the error covariance.
  We agree with the reviewer that this is not an appropriate way to calculate model uncertainty, and in fact we only mention it in our paper to show how inappropriate this approach is. Given that this approach is what is often done to assess uncertainty from ensemble runs (and that instead we propose a different methodology), we prefer to keep this as is in this paragraph. We added a comment to specify that correlations are not considered in this basic technique: "These values quantify, in a rather simple yet often used fashion that neglects any correlation among the ensemble members, the model ensemble-derived uncertainty at the two lidar locations"

- Section 5: This consist of the Machine learning wind speed vertical extrapolation validation. A key question that pops up in the round-robin validation is how correlated the sites are. The results It is not easy to understand if the slight increase in rmse of around 12.5% for the round-robin validation compared to the same-site validation is high or low, as long as there is no analysis of among-site correlation.

We added the following paragraph to section 5:

When interpreting these results, it is important to also consider the correlation existing between the considered data sets. In fact, the $R^2$ coefficient between observed hourly average 140 m wind speed from the closest pairs of lidars is quite high. For example, $R^2 = 0.84$ when considering the NYSERDA E05 and E06 lidars (about 75 km from each other), and $R^2 = 0.88$ between the NYSERDA E06 and Atlantic Shores 06 lidars (about 55 km from each other). These values drop significantly when considering larger distances, so that we have $R^2 = 0.55$ between the observations of the NYSERDA E05 and Atlantic Shores 04 lidars, which are over 145 km apart. Finally, it is important to remember that the Atlantic Shores 04 and 06 lidars do not have any overlapping time in their periods of record, so that their time series can be considered independent from each other. Given these considerations, it is reasonable to expect that the existing correlations between the data sets have an impact on the good generalization skills found here, but only up to a certain level. For example, the remarkably strong generalization skill found between the two NYSERDA lidars is likely connected to a combination of their long period of records and strong autocorrelation. On the other hand, the random forest still performs well when trained and tested at lidars over 140 km apart (the NYSERDA E05 and Atlantic Shores 04 lidars), and even when trained and tested at two lidars (Atlantic Shores 04 and 06) with no overlapping period of records. Therefore, while numbers in Figure 3 are not immune from existing correlations, the overall good generalization performance of the extrapolation algorithm in the relatively limited geographical region considered in our analysis is confirmed.

Finally, we added information on the distance between the lidars in the caption of Figure 1: "The distance between the two NYSERDA lidars is about 75 km, the two Atlantic Shores lidars are about 20 km from each other, and finally the distance between the NYSERDA E05 and Atlantic Shores 04 lidars is about 145 km."

- Section 5: Figure 6 gives the quantification of the relative importance of the various input variables. It is not stated, but I guess this is the random forest out-of-bag estimate? If so, this analysis requires that the features are uncorrelated, so this has to be established before any clear conclusions can be drawn from the analysis. If they are correlated, this should be stated with the possible influences this may have on the results.
  We added the following discussion: "We note that a proper feature importance quantification would require all input variables to be uncorrelated, which is not the case in our analysis. Therefore, the results should be considered as qualitative, and interpreted given the correlation existing between some of the input variables. For example, if the difference between air temperature and SST was not included as input, it is reasonable to expect that the relative importance of air temperature and SST would increase."

- Section 5: The relative importance analysis (fig 6) indicates that only two features account for most of the predictability. If this is the case, one could argue that the machine learning model, could be heavily simplified. This could be investigated, by looking at the quality degradation between a model using all features and one using only the two main features. Using for example AIC (Aike Information Criterion) or other measures that penalizes unnecessary complicated models, one could get an idea if having all features is worth it compared to a simpler model.
  Please see our answer to the next comment.

- Section 5: It is unclear what the benefit of the random forest model is compared to a simple more transparent multilinear regression model? Is the added complexity worth it? It would be very interesting if the authors could provide an estimate of the added value of

the non-linear random forest model compared to a simple multilinear regression model with only two features.

We added the following paragraph to address this aspect:

The fact that only two of the variables used as inputs to the random forest have a relatively large importance introduces the question of whether a simpler algorithm could be used to vertically extrapolate wind speeds. We test this aspect by considering two additional algorithm setups (for simplicity, we only consider the same-site approach), and summarize our results in Table 8:

– First, we test whether a comparable model accuracy can be achieved when considering a random forest that uses only the two most important variables (near-surface wind speed and difference between air temperature and SST) as inputs. We use the same range of hyperparameter and cross-validation setup used in the main analysis. We find that while the model does not overfit the data in a significant way, it has lower skills, with testing RMSE values between 0.15 - 0.40 $\mathrm{m\,s^{-1}}$ larger than what found with the original random forest setup (Table 6).

– Next, we test whether using a simpler regression algorithm with the whole set of input variables considered in the original random forest can lead to a comparable skill. We consider a multivariate linear regression, and use a ridge algorithm (Hoerl and Kennard, 1970) (`Ridge` in Python's library Scikit-learn) to constrain the multivariate regression. We use the same cross-validation setup used in the main analysis, with the only hyperparameter ($\alpha$) for the algorithm sampled between 0.1 and 10. We find that the model does not overfit the data, but it does provide significantly worse extrapolated winds, with RMSE values over 0.5 $\mathrm{m\,s^{-1}}$ larger than what found with the original random forest setup.

Therefore, we conclude that the random forest model considered in the main analysis is an appropriate choice given the complexity of the task of wind speed vertical extrapolation, despite the limited number of variables showing large values of relative importance (likely due to some correlation effects, as described above). Finally, we note that several constraints have been applied to the complexity of the random forest used in the main analysis, in terms of the hyperparameters listed in Table 4, so that the training of the model can be easily completed on a personal computer and only takes a few minutes.

Table 8. Comparison of training and testing RMSE (in modeling hourly average wind speed at 140 m) at the four lidar data sets when using a random forest with a reduced number of input variables and a multivariate linear regression with the whole set of input variables.

| Lidar data set | Random forest with two inputs | | Multivariate linear regression | |
| --- | --- | --- | --- | --- |
| | Training RMSE ($\mathrm{m\,s^{-1}}$) | Testing RMSE ($\mathrm{m\,s^{-1}}$) | Training RMSE ($\mathrm{m\,s^{-1}}$) | Testing RMSE ($\mathrm{m\,s^{-1}}$) |
| NYSERDA E05 | 1.32 | 1.29 | 1.75 | 1.77 |
| NYSERDA E06 | 1.32 | 1.40 | 1.87 | 1.80 |
| Atlantic Shores 04 | 1.47 | 1.55 | 1.77 | 1.85 |
| Atlantic Shores 06 | 1.55 | 1.72 | 2.08 | 1.88 |

- Section 6: The author states that uncertainty increases as the distance from the lidars, used to train the machine learning model, increases. But this is not incorporated in the extrapolation uncertainty. Instead it may turn up as part of the WRF uncertainty as the model is compared to data which have gradually lower quality as the validation is moving away from the lidar used in extrapolation training. This questions the attribution of uncertainty to the different terms (wrf vs total). A discussion on this would help the readers to reflect on how the different uncertainty terms should be interpreted and the fact that they might not be independent from each other.

Thank you for pointing this out. We added a discussion to make the reader consider this aspect in Section 3.2:

Finally, when estimating model uncertainty from measurements, it is important to remember that the measurements themselves have an uncertainty. In our case, we need to consider both the actual instrumental uncertainty ($\sigma_{\mathrm{obs}}$) and the uncertainty connected to the fact the WRF-modeled wind speed will not be compared to directly observed wind speed, but rather to wind speed that has been vertically extrapolated from near-surface observations, using a less-than-perfect algorithm, which was trained at sites different from the ones it is being applied to ($\sigma_{\mathrm{ML}}$). Both these uncertainty components are passed on to the model. For simplicity, we assume that all three components are not correlated with each other, and add $\sigma_{\mathrm{obs}}$ and $\sigma_{\mathrm{ML}}$ in quadrature to the model uncertainty $\sigma_{\mathrm{WRF}}$ estimated using the steps above, to obtain a total uncertainty quantification (JCGM 100:2008, 2008b):

$$\sigma_{\mathrm{tot}} = \sqrt{\sigma_{\mathrm{WRF}}^2 + \sigma_{\mathrm{obs}}^2 + \sigma_{\mathrm{ML}}^2} \tag{1}$$

As will be detailed in the following sections, we will use constant values across the whole considered region for both the instrumental uncertainty $\sigma_{\mathrm{obs}}$ and the extrapolation-related uncertainty $\sigma_{\mathrm{ML}}$. This means that these two uncertainty components will not directly consider the fact that the data set the WRF-modeled hub height wind speed will be compared to has gradually lower quality as we move away from the lidars used to train the extrapolation algorithm. Therefore, this aspect is folded into the uncertainty quantified by $\sigma_{\mathrm{WRF}}$, which therefore assumes a larger connotation compared to the pure WRF model uncertainty: in fact, only if the WRF-modeled wind speed was compared to the *true* hub height wind speed, $\sigma_{\mathrm{WRF}}$ would be a pure quantification only of the uncertainty in numerically modeling hub height winds. Therefore, while for this proof of concept analysis the assumption of uncorrelated uncertainty components can be considered as sufficiently reasonable, and therefore equation 1 justified, future follow-up analyses could explore the potential correlation between different uncertainty sources to further refine the quantification approach we use here.

In Section 6, we also added a comment to remind the reader about this once again.

- The conclusion would benefit from some critical discussion on the choice of methods, possible limitations, and outstanding research questions.
  We have expanded the conclusions, which now read as follows:

**7  Conclusions**

The National Renewable Energy Laboratory has released a state-of-the-art 21-year wind resource assessment product for all the offshore regions in the United States. Because of its numerical nature, this data set has inherent uncertainty, the quantification of which is of primary importance for stakeholders aiming to use this data set to contribute to offshore wind energy growth. In our analysis, we have shown the limits of quantifying model uncertainty in terms of the variability of a model ensemble, which in our case captured only roughly half of the total model uncertainty. Instead, we recommend leveraging observations

to fully capture NWP model uncertainty. In the absence of long-term observed wind speeds at hub height, we have proposed a methodological pipeline to vertically extrapolate near-surface winds from long-term buoy observations using machine learning. We adopt a random forest model, using a number of atmospheric variables measured near the surface as inputs to the regression algorithm. Our approach was well validated across the mid-Atlantic region, and we showed that using a significantly simpler model (either in terms of the regression algorithm itself, or the number of input variables used) would significantly reduce the accuracy of the extrapolated winds. The total model uncertainty we observed in hub-height hourly wind speed was, on average, just below $3 \mathrm{~m\,s}^{-1}$ (about 30% of the mean observed winds). This number is not negligible, especially considering that wind turbine power production is roughly related to the cube of wind speed, but several opportunities exist to reduce this uncertainty in the future.

This analysis is one of many examples of the synergy between NWP models and observations, which points to the multiple interconnections between the two. A larger number of long-term observations are needed to both quantify and, in the long term, reduce the inherent uncertainty of numerical models. In fact, we observe that the uncertainty in the modeled data increases as we move away from the observational data sets used to train the machine learning algorithm. Having a larger number of sites with available hub-height observations covering a variety of atmospheric conditions would allow for the machine learning model to more accurately represent hub height conditions across a wider region. In this context, the sharing of additional proprietary observational data sets should be considered, and the long-term advantages resulting from better numerical modeling should be kept in mind when assessing the overall balance between costs and benefits of such data-sharing initiatives. In the future, the choice of the learning algorithm as well as of the input variables can be explored in more detail, for example by testing a larger number of regression models than what considered here. Also, a similar analysis can be performed for other offshore regions where both a long-term numerical wind resource assessment product and enough observations to assess uncertainty are available.

**Figures**

- Figure 1: State the difference between diamonds and dots in the caption. Done.

- Figure 3: Consider skipping this. Removed.

- Figure 5 and 7: The scatter plot caption needs a sentence about the color coding indicating density of the data. State that the data shown is hourly averaged wind speeds. Done.

- Figure 5 and 7: Plotting some quantile-quantile values as dots on top of the current plot would make it easier to see any systematic differences in the quantiles.

Done, see example below. We also updated the figure captions accordingly.

[Figure]

**Figure 4.** Scatter plot of observed and machine-learning-predicted 140 m hourly average wind speed at the Atlantic Shores 06 lidar when the learning algorithm is trained at the NYSERDA E06 South lidar. The color shades show density of the data, with darker colors indicating regions with more data. The black dots compare quantiles of the two samples.